# Exploring the Relationships between Children’s Oral Health and Parents’ Oral Health Knowledge, Literacy, Behaviours and Adherence to Recommendations: A Cross-Sectional Survey

**DOI:** 10.3390/ijerph191811288

**Published:** 2022-09-08

**Authors:** Ewelina Chawłowska, Monika Karasiewicz, Agnieszka Lipiak, Mateusz Cofta, Brittany Fechner, Agnieszka Lewicka-Rabska, Agata Pruciak, Karolina Gerreth

**Affiliations:** 1Department of Preventive Medicine, Poznan University of Medical Sciences, 60-781 Poznan, Poland; 2Laboratory of International Health, Department of Preventive Medicine, Poznan University of Medical Sciences, 60-781 Poznan, Poland; 3Department of Hypertension, Angiology and Internal Diseases, Poznan University of Medical Sciences, 61-848 Poznan, Poland; 4Institute of Plant Protection-National Research Institute, Research Centre of Quarantine, Invasive and Genetically Modified Organisms, 60-318 Poznan, Poland; 5Department of Risk Group Dentistry, Chair of Paediatric Dentistry, Poznan University of Medical Sciences, 60-812 Poznan, Poland

**Keywords:** health knowledge, oral behaviours, oral health literacy, dental health, home-based health promotion, early childhood caries, preschool children, parents, recommendations, public health

## Abstract

Although preventable, early childhood caries (ECC) is a burdening condition in all WHO regions, with Poland being one of the most affected countries in Europe. Effective home-based prevention of ECC is available and recommended by several expert bodies. Therefore, we wanted to determine how well parents were following selected oral health recommendations in preschool children. Additionally, we wanted to determine which socioeconomic factors influenced such practices the most, and how parents’ oral health literacy and behaviours affected the oral hygiene and oral health status of their children. A cross-sectional survey involving 2338 parents and caregivers of children from 167 kindergartens was performed. Children’s oral hygiene and oral health status were found to be associated with parents’ education and economic status. They were also strongly influenced by parental health literacy and behaviours. With respect to recommendations for preschoolers, the lowest adherence was found in the use of fluoride toothpaste and its age-appropriate amount, supervision of toothbrushing, and refraining from eating after last toothbrushing. The improvement in these areas could be achieved thanks to the involvement of health professionals such as gynaecologists, midwives, and primary care personnel in efforts aimed at increasing parents’ oral health knowledge and literacy.

## 1. Introduction

There is extensive evidence demonstrating that dental caries is still one of the most prevalent non-communicable diseases and widespread childhood conditions worldwide [1,2]. Untreated dental caries (tooth decay) in permanent teeth is the most common health condition according to the Global Burden of Disease 2017 [3]. Early childhood caries (ECC) is defined as tooth decay in children under six years of age [4,5]. Globally, more than 530 million children are estimated to be suffering from caries of primary dentition [3,5]. According to the World Health Organization, the burden of dental caries affects substantial numbers of children in all WHO regions [5]. In the countries of the European region, tooth decay among 6-year-olds varies considerably, being anywhere from 20% to 90% [5]. There are significant differences in oral health between members of different European Union states, with a markedly worse epidemiological situation in a few Eastern and Central European countries [1,5,6,7,8,9]. The oral health of Polish preschoolers is much worse than that of preschoolers in most other European countries [5,6,10]. According to reports from national oral health monitoring programmes commissioned by the Ministry of Health for the years 2016–2020, the prevalence of dental caries increases substantially with age from 41.1% in 3-year-olds to 85.1% in 7-year-olds [11]. Severe ECC was found in 20.9% of 3-year-olds and in 48.5% of 5-year-olds [11]. The situation among children aged 3–6 years has improved over the last few years, but this improvement has primarily been noticed in the youngest age groups and is not fast enough [11].

It is widely known that caries and its complications may severely impact general health and quality of life [1,5,6,7,10]. However, this disease and its negative consequences can be prevented to a large extent. Apart from treatment, there are several recommended primary prevention methods, both professional and home-based [12,13,14]. Recommendations of the Polish Paediatric Dentistry Association in this area [12,15] reflect relevant recommendations from European, American, and international dental associations and public health organisations [4,13,14,16,17,18]. They advocate that home-based prevention for children aged 3–6 years (preschoolers) should include a cariostatic diet, regular use of dental care services, and toothbrushing twice a day with a pea-sized amount of fluoride toothpaste [4,8,12,15,17,18] using either a manual or an electric toothbrush [15,17]. The amount and type of toothpaste should be adjusted to the child’s age [12,15,17]. It should be noted that oral hygiene practices should not be started later than after the first primary tooth has appeared [15,17]. According to a few recommendations, it is the parent or caregiver who should brush the child’s teeth until he or she reaches the age of 8 years [12,15]. Assisting or supervising toothbrushing practices in children up to the age of 7 years is also recommended [17]. Due to an unfavourable risk–benefit ratio, the use of fluoride mouth wash in home prevention is not recommended in children under 6 years of age [12,15,17]. In Poland, unassisted use of dental floss is recommended no earlier than in children aged 8–10 years [15]. In younger children with primary dentition, flossing should be performed by parents or caregivers [15]. The use of dental water jets is only recommended in Poland in particular situations (e.g., disability, increased risk of gingival bleeding, etc.) but not earlier than in 4- to 5-year-olds [15].

Despite a large body of knowledge on the aetiology and prevention of dental caries, it remains to be a serious public health concern imposing a heavy burden on individuals, families, and communities [4,5,14,19]. The child’s oral health depends in part on genetic factors [20], enamel susceptibility, composition of bacterial plaque, level of colonization by cariogenic bacteria (especially Mutans Streptococci), amount and frequency of sugar consumption [4], and amount and characteristics of saliva. Conversely, adequate oral hygiene and supply of fluoride are important protective factors [5,7,8,17,18]. Still, prevention of tooth decay is not an easy task. It is strongly influenced by several social, behavioural, economic, and environmental factors, i.e., the social determinants of health [1,5,10]. They include, for example, family income, parents’ education, place of residence, mother’s age, number of children in the family, parents’ oral hygiene practices, and parents’ oral health status [6,10,19,21,22]. A factor which strongly affects caries prevalence and severity in children is oral health practices and behaviours of children themselves, and of their families and caregivers [5,23].

The available literature emphasises increasingly often that good health competencies are a prerequisite for health-promoting practices. Conversely, low health literacy, defined as an inability to read, understand, and act on health information [24], is associated with poorer health outcomes and a lower use of prevention [25,26,27,28]. Oral health literacy (OHL) is an important component of health competence [29]. OHL may translate into oral health behaviours [30] and oral health status [29]. Moreover, the OHL of caregivers was found to be associated with the oral health status of their 5-year-old children [31].

As the reduction of caries incidence and prevalence in Polish children is progressing too slowly, we believe there is a need to discover why the Polish epidemiological situation is markedly worse than in other EU countries. We decided to take a closer look at preschoolers in the Wielkopolskie province of Poland. The entire population of 3–6-year-olds in Poland includes 1,556,876 children, 10% of whom inhabit the Wielkopolskie province [32]. To our knowledge, no national monitoring studies were conducted in this region in 2016, and 3-year-olds were only studied in 2017 (n = 100). In the years 2016–2020, the national monitoring programme did not include 5-year-olds [11]. Caries prevalence in this province was estimated to be 21% among 3-year-olds, 77% among 6-year-olds (n = 100, 2018), and 85% among 7-year-olds (n = 80, 2019) [11]. To determine what could drive such a high prevalence, we designed a study aiming to determine:how parents adhered to selected recommendations on oral hygiene in their preschool children;how parents’ caries-preventing behaviours and oral health status influenced their children’s behaviours and status.

## 2. Materials and Methods

The presented survey-based cross-sectional study was performed in February and March 2017 in the Wielkopolskie province among parents and caregivers of preschool children aged 3–6 years. This study was funded by the For Health Foundation for Population and International Health (“Dla Zdrowia” Fundacja na Rzecz Zdrowia Populacyjnego i Międzynarodowego) and was inspired by one of the biggest Polish educational and preventive initiatives, called “Caries-Free Childhood” [33], implemented jointly by the Polish Ministry of Health and eight Polish medical universities. Invitations to participate in the survey were sent to all kindergartens in the province, 167 of which participated and made a questionnaire available to parents or guardians of the attending children.

The questionnaire itself was prepared in the Laboratory of International Health at the Department of Preventive Medicine of the Poznan University of Medical Sciences. Although the resulting tool was self-developed, the questions were based on a literature review of the current recommendations on oral health practices of preschoolers. The questionnaire included 67 questions, most of which were closed-ended, and covered 6 areas: (1) socio-demographic questions (residence, age of parents, number of children in the family, parents’ education, and self-assessed economic status), (2) child’s health status including oral health, (3) use of dental care services including preventive care, (4) oral hygiene, (5) cariostatic diet, and (6) respondents’ sources of knowledge. This article presents the results regarding children’s oral health practices and their relations with socioeconomic factors and parental practices.

Seven thousand paper copies of the questionnaire were prepared, each with a leaflet containing instructions on how to complete the questionnaire. They were sent to all the kindergartens which agreed to take part in the study. Additionally, leaflets with instructions were distributed among the parents who preferred to complete the questionnaire online. All participants were informed that participation in the study was fully voluntary and anonymous, and that the data collected would be used for scientific purposes only. The Ethical Committee of the Poznan University of Medical Sciences confirmed that the study did not constitute a scientific experiment and did not need ethical approval. Informed consent was obtained from study participants. Completing the questionnaire took about 15 min on average. The staff of the kindergartens were responsible for distributing the questionnaires, collecting the completed ones, and sending them by post to the project office.

Statistical analyses of the results were performed in PQStat v1.8.2. To compare multiple independent groups with ordinal scales, we used a Kruskal–Wallis test with a post hoc Dunn–Bonferroni test. The dependence between ordinal scales was tested by Spearman’s rank correlation coefficient test. To test relationships between an ordinal scale and a dichotomous variable, the chi-squared test for trends was used. The dependence between nominal scale variables was tested using Pearson’s chi-squared test or Fisher’s exact test (in the case where Cochran’s assumptions about numerosity were not met). To test selected relationships, effect size was measured using the odds ratio along with a 95% confidence interval, while to test the statistical significance of the relationship, the odds ratio (OR) significance test was used. A significance level of 0.05 was assumed for the statistical analyses performed.

## 3. Results

### 3.1. Characteristics of Study Participants

The questionnaire was initially completed by 2462 parents or caregivers of 3–6-year-old children. After 124 incomplete responses were removed, 2338 responses were included in the analyses. Out of these, 1383 were in paper copies (19.8% of the 7000 paper copies sent out to kindergartens). The remaining 955 responses were completed online. Sociodemographic characteristics of the participants are provided in Table 1. Most of the responses (91.6%) were completed by mothers. As regards place of residence, only 13.3% of the participants resided in the city; the remaining sample reported residing in either a small town or village. Most mothers were between the ages of 25–34 years (52.8%), while most fathers were between the ages of 25–34 years (38.4%) and 35–39 years (34.8%). Most participating families had at least one (27%) or two (55.9%) children. Most mothers held higher education (58.4%) and most fathers secondary (36%) or higher education (40.4%). More than a half of the households reported their economic status as being very good (51.8%) with just less than a half of the households reporting it as average (47.8%). More than a half of the participating children were males (52%) and the rest were females (48%). Most participants reported that their child was in good health (97.9%) with 14.2% reporting a chronic illness, and 8.3% of parents reporting their child as having a special diet. Approximately a half of families reported seeing their child’s first tooth appear between the fifth and eighth month of life (46.8%).

### 3.2. Adherence to Recommendations

We investigated parents’ and caregivers’ reported adherence to selected recommendations of the Polish Paediatric Dentistry Association on oral health practices for preschoolers (Table 2). It follows from our analysis of parents’ responses that 60.7% of their children regularly brushed or had their teeth brushed twice a day. The reported use of age-appropriate toothpaste was almost universal (96.4%). While just 85% of parents used fluoride toothpaste in their children, only 72.7% had started this practice by the age of 3 years as recommended. The age-recommended pea-sized amount of toothpaste was used in 66.9% of children. Virtually all the children had toothbrushes. Most used manual ones (71.6%); the rest used electric toothbrushes. Mouth wash was used by 8.2% of children aged 3–6 years. Regular use of dental floss (2.3%) or dental water jets (0.9%) was also rare.

Only 4.4% of parents reported that they always brushed their children’s teeth, and another 9.7% did that often; the remaining parents did not follow the relevant recommendation. A child was “always supervised” while toothbrushing in 46.9% of cases and “often supervised” in 36%. Only 10.1% of parents reported that they “always corrected” toothbrushing results after their children finished toothbrushing, and 16.1% “often corrected” them. A vast majority of parents did not correct brushing results. The recommendation to prevent a child from eating or drinking anything except water after the evening brushing routine was always implemented by 25.3% and nearly always implemented by 25.5% of parents (see Table 2).

In addition, 22.2% of parents (n = 518) had never gone to a dental visit with their preschool child, whereas 60.1% (n = 1405) had already taken their child to a dentist for the first check-up; 8.8% (n = 207) visited a dentist because of their child’s toothache, and 5.5% (n = 129) did that for yet another reason (in 7 out of 10 cases the reason was caries or suspected caries). Only 1.3% (n = 31) of parents stated that a health professional referred their child to go to a dentist, and 2.1% (n = 48) of parents did not remember the reason for the first dental visit.

### 3.3. Socioeconomic Factors vs. Children’s Oral Health Practices and Status

In our study we also explored the influence of selected socioeconomic factors on the reported adherence to recommendations on oral health practices for children and on children’s reported oral health status. The findings are presented in Table 3 in order of decreasing influence.

Parents’ education was proven to be the strongest determinant. The influence of mother’s education was present in 10 out of 15 researched areas. Mothers with higher levels of education tended to report more frequent toothbrushing (r = 0.15; *p* < 0.0001), administering toothpaste for their children (*p* = 0.0019), and using them in the appropriate amounts (r = −0.14; *p* < 0.0001) for their children. It should be noted, however, that the higher mother’s education was, the less frequent the use of fluoride toothpaste became (*p* = 0.0445). Mothers with higher levels of education also more often reported that they brushed their children’s teeth (r = 0.18; *p* < 0.0001), supervised the brushing (r = 0.07; *p* = 0.001), or corrected the brushing results (r = 0.16; *p* < 0.0001). Their children were less often allowed to eat or drink after the evening toothbrushing (r = −0.21; *p* < 0.0001). It comes as no surprise, then, that they were more likely to assess their children’s oral health status as good/quite good (r = 0.17; *p* < 0.0001) and they less often reported children’s tooth extractions due to caries (*p* < 0.0001). The father’s education was similarly significant in relation to oral hygiene. Except the current use of fluoride toothpaste, all the other practices were associated with father’s education (Table 3).

Self-assessed economic status was associated with as many as seven of the areas we studied, although the significance was weak. The worse the status was, the less often children brushed their teeth (r = −0.09; *p* < 0.0001), the later they started using fluoride toothpaste (r = 0.05; *p* = 0.0158), the more often they tended to use toothpaste for adults (*p* = 0.0102), apply inappropriate amounts of toothpaste (r = 0.05; *p* = 0.0172), and brush teeth unsupervised (r = −0.07; *p* = 0.0004). They also more frequently ate or drank after the evening toothbrushing (r = 0.06; *p* = 0,0032). Finally, their parents’ or caregivers’ assessments of children’s oral health status were generally worse (r = −0.09; *p* < 0.0001).

Parents’ age turned out to be associated with six of the questions we asked about. Mother’s age determined the use of fluoride toothpaste (*p* = 0.0060). The older a mother was, the more often her child brushed his or her teeth (r = 0.06; *p* = 0.0036). The children of older fathers tended to use mouth wash more regularly (*p* = 0.0073). Older parents were more likely to brush their children’s teeth (mothers: r = 0.04; *p* = 0.0377; fathers: r = 0.05; *p* = 0.0105), and their children were less likely to eat or drink after the evening toothbrushing (mothers: r = −0.14; *p* < 0.0001; fathers: r = −0.13; *p* < 0.0001). Older parents also tended to assess their children’s oral health as better (mothers: r = 0.08; *p* = 0.0001; fathers: r = 0.05; *p* = 0.0141).

Place of residence was significantly associated with only two areas. Children living in cities tended to brush their teeth more often (*p* = 0.0426) and had started using fluoride toothpaste earlier (*p* = 0.0463) than rural children.

The number of children in the family, according to our findings, was not associated with adherence to most recommendations. There were two exceptions: more children were weakly associated with parents’ lower tendency to brush their children’s teeth (r = −0.05; *p* = 0.0262) and with a less regular use of dental floss (*p* = 0.0437).

None of the studied variables seemed to determine the type of toothbrush used by a child or the use of dental water jets (Table 3).

### 3.4. Oral Health Practices and Status: Parents/Caregivers vs. Children

This part of our analysis presents self-reported oral health status (of the respondent parent only) and practices in parents/caregivers, in addition to the extent of their transfer to preschool children.

As we can observe (Table 4), a vast majority of parents (88.6%) self-assessed their oral health as good or quite good, although concurrently, 45.8% reported having a tooth extraction due to caries. Visiting a dentist within the last 6 months was reported by 64.5% of respondents. With respect to oral hygiene, 71.5% of participants reported that they always brushed their teeth at least twice daily, mostly with a manual toothbrush (72%). In 96.3% of the families both parents used fluoride toothpaste, but in 2.4% none of them did. The use of mouth wash, dental floss, and dental water jets by both parents was rather uncommon (24.9%, 15.7%, and 2.6%, respectively).

Next, we attempted to determine if reported oral health status and practices in parents/caregivers changed the odds of similar status and practices in children (Table 5). It turned out that when one parent reported having good/quite good oral health, the odds of his/her child having the same reported status were 4.15 times greater than when one parent reported bad/rather bad oral health (OR [95% CI] 4.15 [3.06–5.62]; *p* < 0.0001). An extraction due to caries reported by at least one parent increased the odds of this event in the child 2.57 times (OR [95% CI] 2,57 [1.61–4.09]; *p* < 0.0001). The chances of the children whose parents reported visiting a dentist within the last 6 months were 3.92 times greater than those of the children whose parents had such visits within the last year or earlier (OR [95% CI] 3,92 [3.16–4.87]; *p* < 0.0001). The children whose parent always followed a brush-twice-a-day routine had 9.59 times greater odds of following the same routine than those of the other parents (OR [95% CI] 9.59 [7.69–11.95]; *p* < 0.0001). When both parents reported using fluoride toothpaste, the odds of their child using it increased 17.89 times (OR [95% CI] 17.89 [11.00–29.09]; *p* < 0.0001). The use of an electric toothbrush in parents increased the odds of its use in children 6.83 times (OR [95% CI] 6.83 [5.59–8.35]; *p* < 0.0001). The details of the analysis are presented in Table 5.

We found a significant association between parents and children skipping the routine of brushing teeth twice a day (*p* < 0.0001). These associations are presented in Figure 1. There seems to be a tendency for parents to transfer their consistency with this routine to their children.

We also noted a significant association between the amounts of toothpaste used by parents and their children (*p* < 0.0001; see Figure 2). While a pea-sized amount is recommended for preschoolers, some parents fail to follow this recommendation and tend to use as much toothpaste for children as they use for themselves.

## 4. Discussion

Epidemiological data on dental caries is regularly collected during dental check-ups and assessments made as part of the national monitoring programme and other studies [7,10,11,19]. Thanks to this, we as public health practitioners are aware that the prevalence and severity of dental caries in Polish children is high compared to that of other European countries. A comprehensive analysis of the situation is necessary to identify potential barriers which hamper reducing the scale of the problem.

The need to reverse the negative trends is even more pressing when we consider limited access to clinical dental preventive services [1], the growing cost of dental health services in general, and possible negative effects from the COVID-19 pandemic on children’s oral health [34,35]. Parents and caregivers play a decisive role in monitoring home-based caries prevention (proper eating habits, oral hygiene practices) in addition to the use of dental care services of their preschool children [7]. Thus, these were the behaviours we wanted to focus on in the present study.

### 4.1. Adherence to Recommendations

Earlier in this article we cited selected recommendations which should be communicated to parents and caregivers since their leading role as promoters of healthy behaviours in their children has been demonstrated in numerous studies [10]. It was suggested that healthcare professionals should educate parents on the importance of regular dental visits and home-based prevention to eliminate or reduce the influence of ECC risk factors and strengthen the protective ones [10,15,36]. Moreover, it was proposed that fluoride toothpaste manufacturers, public health institutions, and national societies should provide the public with clear oral hygiene visual instructions on toothpaste packaging and in toothbrushing manuals [17].

In practice, our findings seem to indicate that oral health recommendations targeted at preschool children are not universally followed. One of them (the recommendation to use fluoride toothpaste) is based on a large body of research demonstrating that it is likely the primary factor that contributed to reducing the prevalence of dental caries within the last 40 years [12,17]. Yet, according to our participants’ accounts (Table 2), 27.3% of children started using fluoride toothpaste too late, while 12.6 did not use it at all. Additionally, in 2.4% of families neither parent was using fluoride toothpaste. This finding may be related to a problem of the fear of fluoride compounds [7]. According to a 2017 study, a fifth of parents did not know if the toothpaste used by his/her child contained fluoride, but 27.5% admitted they were using fluoride-free toothpaste [12]. As for regular twice-a-day toothbrushing, a routine which was found to decrease caries risk considerably when compared to toothbrushing irregularly or 1–3 times a week [10], it was followed by 60.7% of children in our study group, with nearly 10% brushing their teeth irregularly once a day or even less often. In the 2017 study only a half of 3-year-olds had their teeth cleaned at least twice a day [12]. The amount of toothpaste used was also a serious problem: as many as 33.1% of our participants indicated using inadequate amounts in their children. Moreover, excessive amounts were applied in 2–6-year-olds by 75% of Polish parents [12]. It appears then that more emphasis should be placed in Poland on informing parents and caregivers how important it is to use age-adequate toothpaste amounts. With respect to the use of mouth wash, dental floss, and dental water jets, our study suggests they are not in regular use in preschoolers. It confirms earlier findings demonstrating that children of that age were unlikely to use dental floss [37].

Expert societies are unanimous in recommending that caregivers should either supervise the use of fluoride toothpaste in preschoolers or even brush their children’s teeth themselves [4]. Despite this, the available research suggests that Polish preschoolers tend to brush their teeth single-handedly [37] and that only 1 child in 3 has his or her teeth cleaned by an adult [12]. In our group, 14.1% of parents stated that they always or often brushed their children’s teeth, while 82.9% said their children did it on their own under adult supervision. However, merely 26.2% of parents always or often corrected brushing results after their children completed the toothbrushing routine. Additionally, half of parents admitted allowing their children to drink or eat after the evening toothbrushing routine at least from time to time.

Another aspect we observed in our study was the use of dental care services. We found that 9.7% of parents had visited a dentist over a year before, and 22.2% had never taken their children to a dentist. It may suggest that the importance of healthy primary dentition is underestimated by parents, perhaps also with children not ready to cooperate [11,19]. As a result, parents may not have a chance to obtain professional relevant information on proper paediatric oral care. A survey performed among European dental professionals sheds some light on the limited availability of educational interventions in dental offices [1]. In theory, the Polish public healthcare system offers free dental care to children, including caries risk assessment [1]. Unfortunately, Poland and Romania are the only two out of 27 EU countries whose public healthcare systems do not provide brief dental interventions involving oral health education and promotion [1]. The education available in Poland covers only basic information (importance of toothbrushing, toothbrush, toothpaste) and supervised toothbrushing. Unfortunately, it does not refer to toothbrushing techniques, tooth cleaning methods or products (dental floss, mouth wash, interdental brush), does not promote fluoride toothpaste or dietary awareness, does not educate parents or pregnant women, and does not inform them about habits harmful for oral health [1]. In our opinion, it is then quite likely that many parents do not know the current recommendations they should be following, which is one of the reasons why adherence to them is not as universal as one could wish. That is why it might be beneficial to encourage other healthcare staff to inform parents of current oral health recommendations in an accessible way. Such an educational role could be performed by those professionals who encounter parents before they visit a paediatric dentist. Before a child is born, midwives and gynaecologists who look after an expectant mother could incorporate oral health messages in prenatal visits. Later, education could be continued and complemented by paediatricians, family doctors, and other primary care personnel. The current situation seems favourable, because primary care physicians tend to be consulted more often than any other doctors [38]. Moreover, a new model of primary care with a bigger focus on disease prevention and care coordination has been tested [39] and is about to be launched [40] in Poland. Thus, this could be a step toward integrating paediatric oral health promotion into overall health care [41].

### 4.2. Socioeconomic Factors vs. Children’s Oral Health Practices and Status

We managed to review certain caries risk factors and list them in the order of importance among Polish preschoolers, a group particularly susceptible to this disease. It was reported earlier that socioeconomic factors still determine the use of dental care services by this population in Poland [19]. We can confirm that some of them, including the mother’s and father’s education, are associated with adherence to home-based caries prevention recommendations. An interesting finding, which needs more in-depth research, was the fact that the more educated a mother was, the less likely a child was to use fluoride toothpaste. A variable that was also associated with child’s oral hygiene practices was self-assessed economic status. Another study performed in a group of 3-year-olds indicated that the mother’s education and family’s economic status were weakly associated with caries severity expressed with the DMFT index [10]. One earlier study reported that parents’ education and family’s economic status influenced attitudes toward preventive visits [19]. The odds ratios for attending yearly dental check-ups despite a lack of pain were 2.5-fold greater in children with at least one parent with higher education compared to those whose neither parent had higher education. Low financial burden of oral health expenditures increased the chances 3-fold [19]. With respect to place of residence, a recent national monitoring report found a higher caries prevalence among 6-year-olds inhabiting villages than among those living in cities [11]. Our results indicated that preschoolers living in an urban residence tended to brush their teeth slightly more often and started using fluoride toothpaste earlier. However, an analysis of socio-epidemiological studies found that within the last three decades the difference in caries severity between children living in urban and rural areas, with the latter being previously more affected, had decreased almost 5 times; similar trends were observed in oral health behaviours [42].

### 4.3. Oral Health Practices and Status: Parents/Caregivers vs. Children

By indicating how parents’ self-assessed oral health status and behaviours were associated with children’s oral health status and behaviours, we wanted to highlight why it is worth conducting the oral health education and promotion recommended by numerous international organisations and research teams cited in this article.

For example, we discovered that when both parents used fluoride toothpaste, it increased the chances for the same routine in children by 16.41 times. One parent sticking to a twice-a-day toothbrushing routine increased the child’s odds by 9.59 times. Children also tended to ‘inherit’ parental choices of toothbrush type and amount of toothpaste used. Self-reported good/quite good oral health status in parents translated into over a 4-fold increase of chances for such a status in children. Our results confirmed similar earlier findings [10]. Other authors reported that frequent dental visits of the parent resulted in an almost 4-fold increase of the chances of similar practices in the child, while both parents visiting a dentist regularly were associated with an increased number of caries-free children [30,36]. The child’s chances of being caries-free were greater when parents self-assessed their oral health as good, whereas caries in child’s primary dentition was associated with parents’ oral health [10].

To promote their children’s oral health, parents themselves need to have adequate health knowledge, attitudes, skills, and practices. They also need to have sufficient health literacy [43]. It is of paramount importance to adjust oral health communication to target an audience’s health literacy so that the messages are clear and easy to understand [30].

Therefore, both individual oral health education and public oral health programmes aimed at parents and caregivers should not simply provide information about current recommendations ‘as is,’ but must do it in such a way as to facilitate understanding, lead to real behaviour change and, consequently, to better health outcomes in both parents and children. Our findings, in addition to the available research, seem to indicate that present health education and promotion efforts are scarce, made ad hoc, and are not effective enough. There is a need to develop and evaluate a new systemic approach to oral health communication targeted at preschoolers’ parents. It should include practical hands-on skill training and consider parents’ health literacy. Apart from the scarcity of available educational interventions in Poland [1], a failure to adjust them to the targets’ health literacy might be the reason why recommendations are not adhered to closely enough. While several tools measuring general and oral health literacy have been developed [25,29,30], it might be useful to develop or adapt a Polish culture- and context-sensitive scale measuring parents’ oral health literacy so as to find out which of its dimensions represent the biggest challenge. We believe it is an important next step and a joint task for dental professionals, public health specialists, and health promoters and educators. We also believe that the results presented here might help in future efforts to develop more effective oral health education and promotion initiatives that could facilitate and accelerate the improvement of preschoolers’ oral health.

### 4.4. Implications for Practice

Home-based caries prevention, as the most cost-effective preventive approach, could help reduce the burden of ECC among Polish preschoolers. The improvement in the area of prevention could be achieved through the involvement of health professionals such as gynaecologists, midwives, and primary care personnel in the efforts aimed at increasing parents’ oral health knowledge and literacy. There is a need for educational interventions directed to medical personnel and to parents. Also, it might be beneficial to adapt or develop a scale to measure oral health literacy among Poles to be used in future public health programmes and national monitoring.

### 4.5. Limitations

To effectively promote universal adherence to oral health recommendations, multi-level comprehensive activities are necessary. Our study explored only a few selected aspects. Our results were not based on dental check-ups or medical records but on self-reported outcomes and parental assessments of certain practices. Thus, the results may be subject to recall bias or reflect parents’ and caregivers’ reluctance to report inadequate behaviours.

Although the sample was relatively big, it may be representative of the Wielkopolskie province alone. In addition, the sample included only those parents and caregivers whose children attended preschool facilities.

## 5. Conclusions

The present study identified suboptimal oral health behaviours among preschool children as well as their parents. Our findings indicate that increasing effectiveness of home-based prevention in children requires:intensifying the efforts of health professionals and public health institutions toward better communication of recommendations targeted at parents;considering that certain socioeconomic factors may continue to influence adherence to recommendations, but certain mistakes in oral health practices are reported by all groups irrespective of residence or number of children, so all parents need adequate support;remembering that oral hygiene practices are strongly determined by relevant parental practices and, consequently, that health interventions aimed at children should always involve parents;

From a public health perspective, focusing on the primary prevention of dental caries should be prioritised and could become the opportunity to divert the negative epidemiological trends we can observe in Central and Eastern European countries, including Poland.

## Figures and Tables

**Figure 1 ijerph-19-11288-f001:**
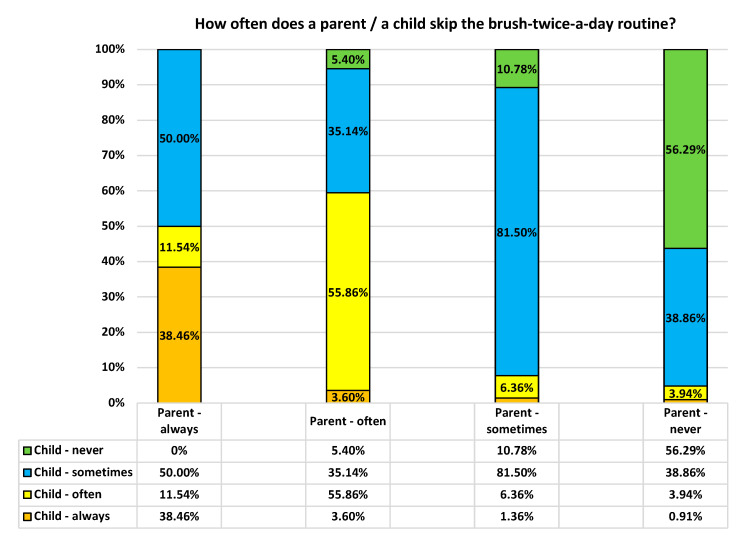
Skipping the twice-a-day toothbrushing routine: parents vs. their children.

**Figure 2 ijerph-19-11288-f002:**
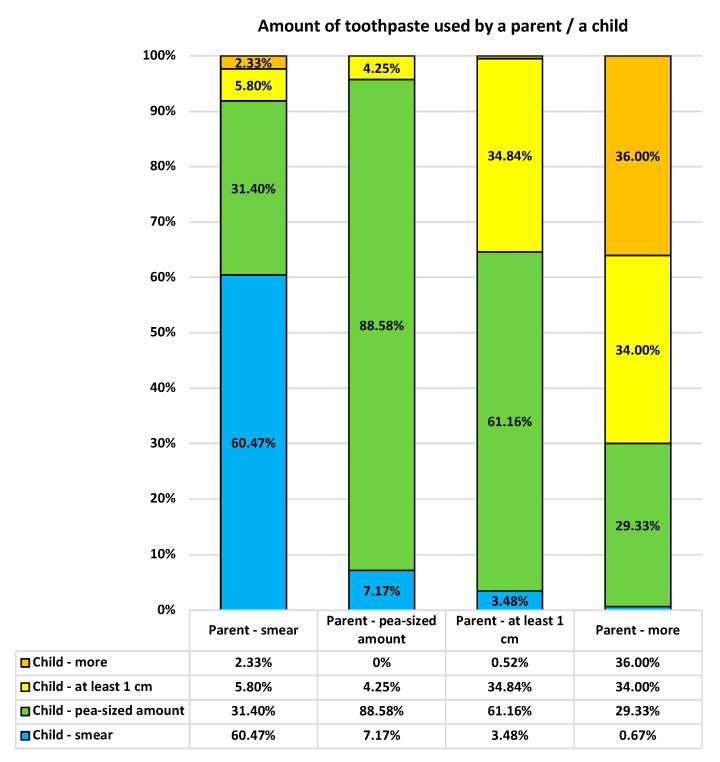
The amounts of toothpaste used: parents vs. their children.

**Table 1 ijerph-19-11288-t001:** Sociodemographic characteristics of study participants.

Sample Characteristics	n	%
Total	2338	100
Place of residence		
Village	975	41.7
Small town (≤50,000 inhabitants)	1052	45.0
City (>50,000 inhabitants)	311	13.3
Mother’s age		
<25	31	1.3
25–34	1235	52.8
35–39	750	32.1
40–44	268	11.5
≥45	54	2.3
Father’s age		
<25	10	0.4
25–34	898	38.4
35–39	813	34.8
40–44	462	19.8
≥45	155	6.6
Number of children in the family		
One	632	27.0
Two	1307	55.9
Three or more	399	17.1
Mother’s education		
Primary	33	1.4
Vocational	205	8.8
Secondary	735	31.4
Higher	1365	58.4
Father’s education		
Primary	58	2.5
Vocational	494	21.1
Secondary	842	36.0
Higher	944	40.4
Economic status		
Very good	1211	51.8
Average	1118	47.8
Bad/Very bad	9	0.4
Child’s gender		
Boy	1216	52.0
Girl	1122	48.0
Child’s general health status		
Good	2288	97.9
Not good	43	1.8
Unknown	7	0.3
Child’s chronic illness		
Yes	332	14.2
No	2006	85.8
Child’s special diet		
Yes	194	8.3
No	2144	91.7
First tooth (month)		
1–2	13	0.6
3–4	226	9.6
5–8	1095	46.8
9–11	243	10.4
>12	76	3.3
I don’t remember	685	29.3

**Table 2 ijerph-19-11288-t002:** Oral health practices in children according to parents/caregivers.

Oral Health Practices in Children		Results n = 2338	
		n	%
**Frequency and regularity of toothbrushing**	Always twice a day	1419	**60.7**
	Twice a day, not always	687	29.4
	Once a day, not always	213	9.1
	Less often	19	0.8
**Type of toothpaste used**	The same toothpaste as everyone else in the family	83	3.6
	Toothpaste for children adequate for their age	2255	**96.4**
**Current use of fluoride toothpaste**	Yes	1987	**85**
	No	351	15
**Age at which fluoride toothpaste was introduced**	Between 0–1 year	281	**12**
	Between 1–2 years	770	**32.9**
	Between 2–3 years	649	**27.8**
	Later	343	14.7
	Child does not use fluoride toothpaste	295	12.6
**Amount of toothpaste used**	A smear	154	6.5
	Not more than a pea-sized amount (around 0.5 cm)	1563	**66.9**
	At least 1 cm	558	23.9
	More	63	2.7
**Type of toothbrush used**	Manual	1674	**71.6**
	Electric	663	**28.4**
	Child does not have a toothbrush	1	0
**Use of mouth wash**	Yes, regularly	191	8.2
	No	2147	**91.8**
**Use of dental floss**	Yes, regularly	54	2.3
	No	2284	**97.7**
**Use of dental water jet**	Yes, regularly	20	**0.9**
	No	2318	**99.1**
**Parent/caregiver brushing child’s teeth**	No, never	701	30
	Hardly ever	429	18.4
	Sometimes	877	37.5
	Often	227	9.7
	Always	104	**4.4**
**Adult supervising child’s toothbrushing**	No, never	29	1.2
	Hardly ever	34	1.4
	Sometimes	338	14.5
	Often	841	36
	Always	1096	**46.9**
**Parent/caregiver correcting brushing results**	No, never	394	16.9
	Hardly ever	299	12.7
	Sometimes	1033	44.2
	Often	376	16.1
	Always	236	**10.1**
**Child eating or drinking after the evening toothbrushing**	No, never	592	**25.3**
	Hardly ever	596	25.5
	Sometimes	940	40.2
	Often	165	7.1
	Always	45	1.9

Note: percentages in bold indicate the parents/caregivers who adhered to relevant recommendations.

**Table 3 ijerph-19-11288-t003:** Selected socioeconomic factors vs. children’s oral health practices and status according to parents/caregivers.

	Mother’s Education	Father’s Education	Self-Assessed Economic Status	Mother’s Age	Father’s Age	Place of Residence	Number of Children in the Family
**Frequency and regularity of toothbrushing**	**r = 0.15** ** *p* ** **< 0.0001** ^a^	**r = 0.16** ** *p* ** **< 0.0001** ^a^	**r = −0.09** ** *p* ** **< 0.0001** ^a^	**r = 0.06** ** *p* ** **= 0.0036** ^a^	r = 0.03 *p* = 0.1956 ^a^	*p* = 0.0426 ^c^	r = −0.04 *p* = 0.0789 ^a^
**Type of toothpaste used**	** *p* ** **= 0.0019** ^b^	** *p* ** **= 0.0014** ^b^	** *p* ** **= 0.0102** ^b^	*p* = 0.8117 ^b^	*p* = 0.0511 ^b^	*p* = 0.0552 ^d^	*p* = 0.5256 ^b^
**Current use of fluoride toothpaste**	** *p* ** **= 0.0445** ^b^	*p* = 0.1471 ^b^	*p* = 0.8682 ^b^	** *p* ** **= 0.0060** ^b^	*p* = 0.0694 ^b^	*p* = 0.4441 ^d^	*p* = 0.2509 ^b^
**Age at which fluoride toothpaste was introduced**	r = −0.01 *p* = 0.5558 ^a^	r = −0.03 *p* = 0.2093 ^a^	**r = 0.05** ** *p* ** **= 0.0158** ^a^	r = 0.03 *p* = 0.1448 ^a^	r = 0.03 *p* = 0.1629 ^a^	** *p* ** **= 0.0463** ^c^	r = −0.01 *p* = 0.7494 ^a^
**Amount of toothpaste used**	**r = −0.14** ** *p* ** **< 0.0001** ^a^	**r = −0.08** ** *p* ** **= 0.0002** ^a^	**r = 0.05** ** *p* ** **= 0.0172** ^a^	r = 0.01 *p* = 0.7591 ^a^	r = 0.01 *p* = 0.7132 ^a^	*p* = 0.2153 ^c^	r = −0.02 *p* = 0.3173 ^a^
**Type of toothbrush used**	*p* = 0.1503 ^b^	*p* = 0.0694 ^b^	*p* = 0.1182 ^b^	*p* = 0.7732 ^b^	*p* = 0.8378 ^b^	*p* = 0.7868 ^d^	*p* = 0.0703 ^b^
**Use of mouth wash**	*p* = 0.8633 ^b^	*p* = 0.4526 ^b^	*p* = 0.0567 ^b^	*p* = 0.3312 ^b^	** *p* ** **= 0.0073** ^b^	*p* = 0.1328 ^d^	*p* = 0.3025 ^b^
**Use of dental floss**	*p* = 0.7382 ^b^	*p* = 0.906 ^b^	*p* = 0.8362 ^b^	*p* = 0.8998 ^b^	*p* = 0.8386 ^b^	*p* = 0.1793 ^d^	** *p* ** **= 0.0437** ^b^
**Use of dental water jet**	*p* = 0.9102 ^b^	*p* = 0.1143 ^b^	*p* = 0.4472 ^b^	*p* = 0.7516 ^b^	*p* = 0.5860 ^b^	*p* = 0.5521 ^e^	*p* = 0.9981 ^b^
**Parent/caregiver brushing child’s teeth**	**r = 0.18** ** *p* ** **< 0.0001** ^a^	**r = 0.15** ** *p* ** **< 0.0001** ^a^	r = −0.04 *p* = 0.068 ^a^	**r = 0.04** ** *p* ** **= 0.0377** ^a^	**r = 0.05** ** *p* ** **= 0.0105** ^a^	*p* = 0.1069 ^c^	**r = −0.05** ** *p* ** **= 0.0262** ^a^
**Adult supervising child’s toothbrushing**	**r = 0.07** ** *p* ** **= 0.001** ^a^	**r = 0.05** ** *p* ** **= 0.0163** ^a^	**r = −0.07** ** *p* ** **= 0.0004** ^a^	r = 0.002 *p* = 0.904 ^a^	r = −0.03 *p* = 0.2107 ^a^	*p* = 0.1293 ^c^	r = −0.04 *p* = 0.0796 ^a^
**Parent/caregiver correcting brushing results**	**r = 0.16** ** *p* ** **< 0.0001** ^a^	**r = 0.15** ** *p* ** **< 0.0001** ^a^	r = −0.04 *p* = 0.0891 ^a^	r = 0.02 *p* = 0.301 ^a^	r = 0.02 *p* = 0.2607 ^a^	*p* = 0.3817 ^c^	r = −0.03 *p* = 0.1625 ^a^
**Child eating or drinking after the evening toothbrushing**	**r = −0.21** ** *p* ** **< 0.0001** ^a^	**r = −0.21** ** *p* ** **< 0.0001** ^a^	**r = 0.06** ** *p* ** **= 0.0032** ^a^	**r = −0.14** ** *p* ** **< 0.0001** ^a^	**r = −0.13** ** *p* ** **< 0.0001** ^a^	*p* = 0.4474 ^c^	r = −0.01 *p* = 0.6134 ^a^
**Child’s oral health status assessed by parents/caregivers**	**r = 0.17** ** *p* ** **< 0.0001** ^a^	**r = 0.19** ** *p* ** **< 0.0001** ^a^	**r = −0.09** ** *p* ** **< 0.0001** ^a^	**r = 0.08** ** *p* ** **= 0.0001** ^a^	**r = 0.05** ** *p* ** **= 0.0141** ^a^	*p* = 0.3942 ^c^	r = −0.02 *p* = 0.3787 ^a^
**Child’s past extractions due to caries**	** *p* ** **< 0.0001** ^b^	** *p* ** **< 0.0001** ^b^	*p* = 0.8879 ^b^	*p* = 0.2348 ^b^	*p* = 0.5714 ^b^	*p* = 0.7647 ^d^	*p* = 0.5546 ^b^

Note: ^a^ Spearman’s rank correlation test; ^b^ Chi-squared test for trends; ^c^ Kruskal-Wallis ANOVA; ^d^ Pearson’s chi-squared test; ^e^ Fisher’s exact test. Numbers in bold indicate statistically significant results.

**Table 4 ijerph-19-11288-t004:** Reported oral health status and practices in parents/caregivers.

Oral Health Status and Practices in Parents/Caregivers		Results n = 2338	
		n	%
**Oral health status**	Good/Quite good	2072	88.6
	Bad/Rather bad	249	10.7
	I don’t know	17	0.7
**Past extractions due to caries**	Yes	1070	45.8
	No	1263	54
	No answer	5	0.2
**Last dental visit**	Within the last 6 months	1508	64.5
	Within the last 12 months	604	25.8
	Earlier	226	9.7
**Toothbrushing twice a day**	Never	25	1.1
	Sometimes	229	9.8
	Often	412	17.6
	Always	1672	71.5
**Use of fluoride toothpaste**	By both parents	2251	96.3
	Only by mother	14	0.6
	Only by father	18	0.7
	Not used by parents	55	2.4
**Type of toothbrush used**	Manual	1683	72
	Electric	655	28
	Parent does not have a toothbrush	0	0
**Use of mouth wash**	By both parents	582	24.9
	Only by mother	208	8.9
	Only by father	223	9.5
	Not used by parents	1325	56.7
**Use of dental floss**	By both parents	368	15.7
	Only by mother	380	16.3
	Only by father	124	5.3
	Not used by parents	1466	62.7
**Use of dental water jet**	By both parents	60	2.6
	Only by mother	38	1.6
	Only by father	17	0.7
	Not used by parents	2223	95.1

**Table 5 ijerph-19-11288-t005:** Oral health status and practices: parents/caregivers vs. children.

**Reported Oral Health Status**		**Child**		**OR [95% CI]** ** *p* **
		Good/Quite good	Bad/Rather bad	
**Respondent parent/caregiver**	Good/Quite good	1845 (89.65%)	213 (10.35%)	4.15 [3.06–5.62]; <0.0001
	Bad/Rather bad	165 (67.62%)	79 (32.38%)	
**Past extractions due to caries**		**Child**		**OR [95% CI]** ** *p* **
		Yes	No	
**Respondent parent/caregiver**	Yes	57 (5.33%)	1013 (94.67%)	2.57 [1.61–4.09]; <0.0001
	No	27 (2.14%)	1232 (97.86%)	
**Last dental visit**		**Child**		**OR [95% CI]** ** *p* **
		Within the last 6 months	Within the last 12 months or earlier	
**Respondent parent/caregiver**	Within the last 6 months	1015 (80.94%)	239 (19.06%)	3.92 [3.16–4.87]; <0.0001
	Within the last 12 months or earlier	301 (51.99%)	278 (48.01%)	
**Toothbrushing at least twice a day**		**Child**		**OR [95% CI]** ** *p* **
		Always	Not always	
**Respondent parent/caregiver**	Always	1152 (68.90%)	520 (31.10%)	9.59 [7.69–11.95]; <0.0001
	Not always	125 (18.77%)	541 (81.23%)	
**Use of fluoride toothpaste**		**Child**		**OR [95% CI]** ** *p* **
		Yes	No	
**Both parents/caregivers**	Yes	1963 (87.21%)	288 (12.79%)	17.89 [11.00–29.09]; <0.0001
	No	24 (27.59%)	63 (72.41%)	
**Type of toothbrush used**		**Child**		**OR [95% CI]** ** *p* **
		Electric	Manual	
**Respondent parent/caregiver**	Electric	380 (58.02%)	275 (41.98%)	6.83 [5.59–8.35]; <0.0001
	Manual	283 (16.83%)	1399 (83.17%)	

Note: A number smaller than n = 2338 indicates exclusion of “I don’t know” responses, and cases when children had not been to a dental visit yet or did not have a toothbrush.

## Data Availability

Not applicable.

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
