# Peer review of "Exploring the Relationships between Children’s Oral Health and Parents’ Oral Health Knowledge, Literacy, Behaviours and Adherence to Recommendations: A Cross-Sectional Survey"

_ijerph, 2022, doi:10.3390/ijerph191811288_

Round 1
Reviewer 1 Report
I have reviewed with great interest your work entitled ”Early childhood caries prevention starts at home. Exploring the relationships between children’s oral health and parents’ oral health knowledge, literacy, behaviours and adherence to recommendations: A cross-sectional survey”.
I congratulate the authors for all the effort that they did to conduct this study.
The topic is very interesting and appreciable from a scientific and clinical point of view. The study is well designed and conducted, and the manuscript is clear.
The introduction is well organized and clear.
The methodology is well described and complete.
The results are clear and supported by the increased numbers of tables. The statistical analysis is correct.
The discussion is well conducted and justified by the articles from literature.
The conclusions are correct.
Additional Comments:
The research team applied a structured original questionnaire to the participants in the study, consisting of six sections: (1) a socio-demographic questions, (2) a child’s health status including oral health, (3) use of dental care services including preventive care, (4) oral hygiene, (5) cariostatic diet, and (6) respondents’ sources of knowledge.
The study provides a detailed analysis regarding the factors which are involved in oral health status of the children from Poland.
In my viewpoint, I do not have any comments regarding the methodology of this important issue covered in this study.
I observed that the text of article has a few grammatical errors. A part of the sentences are too long and it is difficult for the readers to understand them. Sometimes, from a grammatical point of view, the structure of the sentence/syntagma is not correct.
The results are summarized by the increased numbers of tables.
The authors should reduce the length of the conclusion, which should only report the main findings of the study.
Also, the reference format should be standardized.
Reviewer 2 Report
Thank you for the opportunity to review your study. the study certainly presents a very significant sample size and allows an assessment of habits and propensity for a correct management of oral health in the geographical area of reference, highlighting the critical issues on which to work.
some minor revisions are required:
in the M&M section, specify better the type of administration of the questionnaire.
Who provided the instruction to the parents / guardians who filled out the paper questionnaire?
Line 125 specify the age range of the participating children.
In the results add the data on the percentage of participants compared to the initial expected sample.
I would not mention ECC in the title as the study does not report clinical data and could be misleading to the reader.
A minor revision of the English and text is required.
Reviewer 3 Report
I think that there should be a multivariate analysis as it would be expected that higher level of education, family income and age of becoming a parent could be related - this will not be found by testing the variables soarately
